# Empowering Power Outage Prediction with Spatially Aware Hybrid Graph Neural Networks and Contrastive Learning

**Xuyang Shen**                                                  *xuyang.shen@uconn.edu*
*University of Connecticut*

**Zijie Pan**
*University of Connecticut*

**Diego Cerrai**
*University of Connecticut*

**Xinxuan Zhang**
*Western Connecticut State University*

**Christopher Colorio**
*Eversource Energy*

**Emmanouil N. Anagnostou**
*University of Connecticut*

**Dongjin Song**                                                 *dongjin.song@uconn.edu*
*University of Connecticut*

**Reviewed on OpenReview:** *https://openreview.net/forum?id=Vf5FDYrOiU*

## Abstract

Extreme weather events, such as severe storms, hurricanes, snowstorms, and ice storms, which are exacerbated by climate change, frequently cause widespread power outages. These outages halt industrial operations, impact communities, damage critical infrastructure, profoundly disrupt economies, and have far-reaching effects across various sectors. To mitigate these effects, the University of Connecticut and Eversource Energy Center have developed an outage prediction modeling (OPM) system to provide pre-emptive forecasts for electric distribution networks before such weather events occur. However, existing predictive models in the system do not incorporate the spatial effect of extreme weather events. To this end, we develop Spatially Aware Hybrid Graph Neural Networks (SA-HGNN) with contrastive learning to enhance the OPM predictions for extreme weather-induced power outages. Specifically, we first encode spatial relationships of both static features (*e.g.*, land cover, infrastructure) and event-specific dynamic features (*e.g.*, wind speed, precipitation) via Spatially Aware Hybrid Graph Neural Networks (SA-HGNN). Next, we leverage contrastive learning to handle the imbalance problem associated with different types of extreme weather events and generate location-specific embeddings by minimizing intra-event distances between similar locations while maximizing inter-event distances across all locations. Thorough empirical studies in four utility service territories, *i.e.*, Connecticut, Western Massachusetts, Eastern Massachusetts, and New Hampshire, demonstrate that SA-HGNN can achieve state-of-the-art performance for power outage prediction.

# 1    Introduction

Power outages caused by severe weather events, such as hurricanes, snowstorms, and heavy rainfall, pose significant risks to modern society by disrupting critical infrastructure and essential services across sectors like healthcare, transportation, and finance. In the United States, weather-related power outages cost the economy an estimated $18 to $70 billion annually, with the frequency and severity of billion-dollar disasters steadily increasing over the past two decades  (Campbell, 2012; Smith, 2020). These outages can result in significant economic losses and, in extreme cases, loss of life  (Flores et al., 2022; Dominianni et al., 2018). For instance, over 15 months between 2011 and 2012, three major storms in Connecticut caused extensive outages, affecting hundreds of thousands of people and resulting in considerable economic losses. Similarly, Hurricane Sandy inflicted severe damage on New York, causing prolonged service disruptions and operational challenges for utility companies  (Yang et al., 2020; Udeh et al., 2024). Therefore, accurate forecasting of the magnitude and spatial distribution of weather-induced power outages is essential to mitigate these impacts. Such forecasts can inform evacuation strategies, improve storm response planning, and guide investments in reinforcing and upgrading electrical infrastructure  (Baembitov et al., 2023; Cerrai et al., 2019a;b; D'Amico et al., 2019).

Although outage prediction has received increasing attention recently, existing approaches still suffer from critical limitations. Traditional machine learning methods, such as ensemble models, have shown promising results by improving prediction accuracy and robustness  (Nateghi et al., 2014; Wanik et al., 2015; He et al., 2017; Cerrai et al., 2019a;c; Udeh et al., 2024). However, they treat each location independently and do not explicitly model spatial relationships between geographic regions, which are crucial for understanding the spatial effect of storm impacts. Although convolutional and recurrent neural networks, including their advanced variants  (Han et al., 2022; Sun et al., 2022), can capture spatiotemporal dependencies in grid-structured data such as radar images or weather maps, their dependency on rigid Euclidean grids limits their applicability to sensor networks or outage datasets defined on irregular geographic layouts. More recently, graph-based approaches have emerged as powerful tools for modeling non-Euclidean spatial dependencies, allowing each node to represent a spatial location. Nevertheless, existing GNN-based methods  (Kipf & Welling, 2016; Hamilton et al., 2017a; Owerko et al., 2018; Defferrard et al., 2016) often consider only fixed spatial structures and lack the flexibility to model event-specific spatial dynamics. Furthermore, most prior work overlooks the inherent imbalance in outage datasets, where low-impact events dominate and high-impact events, though rare, carry greater operational significance and value. These gaps motivate the development of a more spatially adaptive and representation-discriminative approach for power outage prediction.

To this end, we propose Spatially Aware Hybrid Graph Neural Networks (SA-HGNN) that leverage contrastive learning to enhance outage prediction models (OPM) for extreme weather-induced power outages. Specifically, we first construct a fixed adjacency matrix to encode the spatial relationships of static features and design a dynamic graph learning module to capture and infer complex, evolving spatial dependencies across different events. We then develop SA-HGNN to integrate spatial dependencies derived from both static and dynamic features. To address the imbalance issue associated with extreme weather events of varying severity, we incorporate a contrastive learning strategy to generate location-specific embeddings. These embeddings are obtained by minimizing intra-event distances between similar locations while maximizing inter-event distances across all locations, resulting in more discriminative representations for each location. Our main contributions include:

- We introduce SA-HGNN, a novel graph-based deep learning model that can effectively integrate both static and dynamic spatial dependencies to enhance outage prediction for extreme weather events.

- We develop a dynamic graph learning module that can capture and infer complex, evolving spatial relationships across different locations, addressing the limitations of existing methods that rely solely on fixed spatial structures.

- To tackle the imbalance issue in outage datasets, we adopt a contrastive learning strategy that learns location-specific embeddings by minimizing intra-event distances between similar locations while maximizing inter-event distances across all locations.

- Our studies in four utility service territories, *i.e.*, Connecticut, Western Massachusetts, Eastern Massachusetts, and New Hampshire, demonstrate that SA-HGNN can achieve state-of-the-art performance for power outage prediction.

## 2 Related work

In recent years, machine learning methods have been increasingly employed to address the challenges of forecasting weather-related power outages (Watson et al., 2022; Garland et al., 2023). One widely adopted approach is ensemble learning. For instance, (Nateghi et al., 2014) utilized Random Forest (RF) to predict outages caused by tropical cyclones, while (Wanik et al., 2015) employed tree-based models to forecast power outages in the New England region. He and Cerrai leveraged Bayesian additive regression trees (BART) to predict outages resulting from storm events (He et al., 2017; Cerrai et al., 2019a;c), whereas (Udeh et al., 2024) explored RF for predicting storm-induced power outages in the New York State region. These ensemble approaches help mitigate overfitting and enhance prediction accuracy by leveraging diverse decision paths and combining the outputs of multiple decision trees. However, they do not explicitly incorporate or exploit spatial information, which is critical for outage prediction as extreme weather events (*e.g.*, heavy rainfall, snowstorms) typically have strong spatial effects.

Convolutional and recurrent neural networks (CNNs and RNNs), including advanced hybrid architectures like ConvLSTM, have been widely adopted to capture spatiotemporal relationships in targeted datasets, including radar images and weather sequences. These models are particularly effective at capturing both spatial and temporal dependencies, which are critical for applications such as short-term weather forecasting. For instance, (Han et al., 2022) demonstrated the use of a U-Net model for convective precipitation prediction, while (Sun et al., 2022) applied a 3D-ConvLSTM model for storm nowcasting, showcasing the strengths of these architectures in handling complex weather data. However, CNNs are inherently designed for processing data with a regular rigid structure, such as images and time series. This makes them less suited for sensor network data used in weather and outage monitoring, with their utility in outage prediction still underexplored.

Recently, the development of graph neural networks (Kipf & Welling, 2016; Hamilton et al., 2017a) provides promising solutions to incorporate the complex spatial relationships of outage data at different locations, where each node corresponds to a spatial location and contains both static features (*e.g.*, land cover, infrastructure) as well as dynamic features (wind speed, precipitation, *etc*). (Owerko et al., 2018) explored various GNN architectures, including ChebNet (Defferrard et al., 2016), to predict weather-induced power outages. More broadly, recent graph-based multitask learning studies have shown that modeling task affinity can improve representation learning and knowledge sharing across related graph prediction problems, including through higher-order task interactions and gradient-based task affinity estimation (Li et al., 2023; 2024). These works strengthen the connection between outage forecasting and the broader graph machine learning literature. However, existing techniques still often fall short in simultaneously integrating static and dynamic features while modeling event-specific spatial dependencies across diverse locations.

In addition, recent advances in contrastive learning have shown promise in improving robustness under data imbalance and distribution shifts. For example, Correct-N-Contrast (Zhang et al., 2022a) encourages representations that distinguish true predictive signals from spurious correlations. This perspective is relevant to outage prediction, where rare but high-impact events may induce substantial imbalance and spurious patterns. However, such methods are not specifically designed for structured spatiotemporal outage forecasting with dynamic spatial dependencies.

## 3 Problem statement

We aim to predict location-specific power outage counts during extreme weather events within a given service territory, i.e., different regions in New England. For each specific territory, we observe a collection of $m$ historical extreme weather events. In every event, a fixed set of $N$ geographical locations is monitored, and each location is described by a feature vector that combines both static and dynamic attributes. Static features capture time-invariant location characteristics, such as topography, land cover, vegetation,

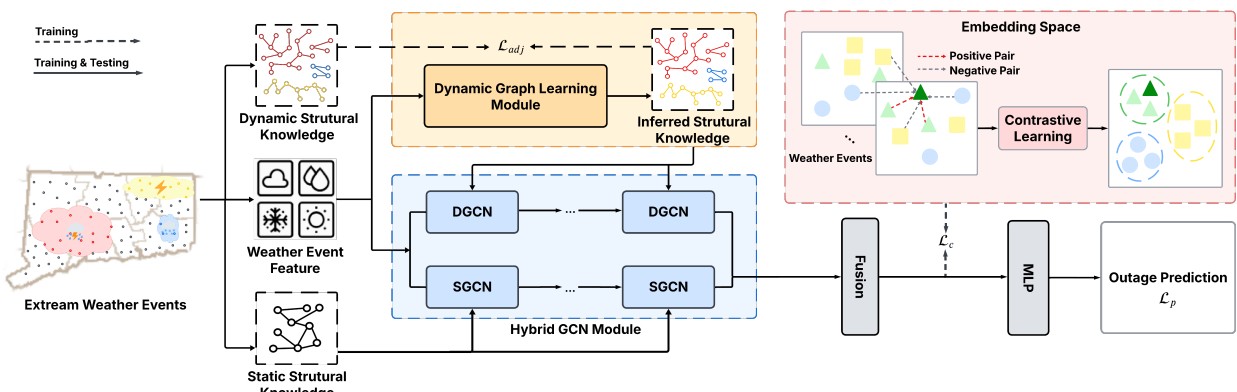

Figure 1: The framework of SA-HGNN. The dynamic graph learning module learns event-specific adjacency matrices guided by external structure, which are used in the dynamic graph convolution. The hybrid graph convolution includes a dynamic GCN (DGCN) for capturing event-specific patterns and a static GCN (SGCN) that aggregates information from a shared graph. Their outputs are concatenated to form location-wise embeddings. Contrastive learning further refines these embeddings by aligning similar locations within events and separating dissimilar ones across events. A regression module then projects the fused embeddings to predict outage values.

and infrastructure properties, which remain fixed across events. Dynamic features capture event-specific meteorological conditions, such as wind speed, precipitation, temperature, and soil moisture observed during a particular extreme weather event. The complete feature description is provided in Appendix A. This separation enables event conditioned representation learning by decoupling persistent spatial context from transient weather-driven effects.

Formally, for an event $k$, let $X_k \in \mathbb{R}^{N \times F}$ denote the input feature matrix, where each row corresponds to a location and $F$ is the number of features per location. The corresponding outage count vector is denoted by $Y_k \in \mathbb{R}^N$, where $Y_{k,i}$ represents the observed outage count at location $i$ during event $k$. Collectively, the data for a territory can be viewed as an event-wise tensor $X \in \mathbb{R}^{m \times N \times F}$. Each event is modeled as a graph $G_k = (V, E_k)$, where the node set $V = \{v_1, \ldots, v_N\}$ corresponds to the fixed locations, and edges represent spatial or functional dependencies between locations, and $E_k$ defines the edges based on spatial or functional relationships. During training, we have access to event-specific adjacency matrices $A_k \in \mathbb{R}^{N \times N}$ constructed from external structural knowledge, which are used to guide the learning of spatial relationships. At inference time, however, the adjacency matrix is not assumed to be known; instead, the model infers an event-specific graph structure $\hat{A}_k$ directly from the input features $X_k$.

Given an event-level input $X_k$, a graph neural network learns spatially informed node representations and predicts outage counts as

$$\hat{Y}_k = f_{\text{GNN}}(X_k, \hat{A}_k; \theta), \tag{1}$$

where $\hat{A}_k$ denotes the learned adjacency matrix inferred from $X_k$, and $\theta$ represents the learnable model parameters. The training objective is to minimize the discrepancy between the predicted outage counts $\hat{Y}_k$ and the ground-truth observations $Y_k$ across events.

## 4 Methodology

In this section, we introduce the general framework of our proposed model Spatially Aware Hybrid Graph Neural Networks (SA-HGNN) with dynamic graph inference and contrastive learning in detail. The overview of SA-HGNN is illustrated in Figure 1. SA-HGNN contains three key components: Dynamic Graph Learning, Hybrid Graph Convolutional Module, and Contrastive Learning Module. To capture latent spatial relationships among locations across different events under dynamically evolving extreme weather conditions, the Dynamic Graph Inference Module (Subsection 4.1) learns and infers a dynamic adjacency matrix for each event, incorporating external structural knowledge to guide the learning process. To incorporate both static and dynamic features, we design a Hybrid Graph Convolutional Module (Subsection 4.2) that

separately processes learned static and dynamic neighbor information through two distinct branches, enabling the model to capture more meaningful spatial dependencies and adapt to varying weather-induced outage patterns. Lastly, the Contrastive Learning Module (Subsection 4.3) is designed to overcome the imbalance issue, enhance generalization, and improve the model's ability to distinguish outage patterns across different extreme weather events. By incorporating intra-event and inter-event contrastive learning, the module refines node embeddings by ensuring that locations with similar outage behaviors under the same weather event are closely aligned, while those experiencing different outage impacts across events remain distinct. This distinction is crucial for capturing the variability of outage patterns under different weather conditions, allowing the model to better adapt to unseen events. Projection layers are applied to the learned node embeddings to the desired output dimension for the final outage predictions. The following subsections provide details of these three key components.

### 4.1 Dynamic Graph Learning Module

For extreme weather events, spatial dependencies among locations vary significantly across events due to the dynamic and localized nature of weather conditions. Relying solely on a fixed, pre-defined graph constructed from static geographic proximity or infrastructure information is therefore insufficient, as such relationships may fail to reflect event-specific interactions induced by extreme weather. To accurately model outage patterns, it is crucial to account for both static relationships, which are driven by persistent geographic and infrastructural characteristics, and dynamic spatial interactions that emerge uniquely under each event. These dynamic interactions may arise from shared exposure to high wind fields, correlated soil moisture conditions, or aligned storm trajectories, and play a key role in shaping event-level outage propagation (Liu et al., 2023; Ye et al., 2022; Zhang et al., 2024).

Inspired by prior work on adaptive adjacency learning (Wu et al., 2020; Shi et al., 2019; Wu et al., 2019; Bai et al., 2020; Chen et al., 2021), we introduce a dynamic graph learning module that constructs an event-specific adjacency matrix conditioned on observed node features. This formulation avoids costly and manual graph construction, which would otherwise require event-specific engineering and detailed domain knowledge for each extreme weather scenario. Instead, the learned adjacency enables the model to adaptively infer spatial dependencies that generalize across unseen events within the same service territory. During training, the learned adjacency is further guided by external structural knowledge (Pan et al., 2024), which provides a weak prior on spatial dependencies and regularizes the graph inference process (see Section 5.1.1 for details).

This module dynamically adjusts the graph structure based on the unique weather features of each event, capturing evolving spatial dependencies that are critical for accurately modeling outage patterns and enhancing prediction performance. Formally, the adjacency matrix $\hat{A}_k$ for each event $k$ is learned via the following formulation:

$$\hat{A}_k = \text{SoftMax}\left(\tanh(X_k \mathbf{W}_1) \cdot \tanh(\mathbf{W}_2^\top X_k^\top)\right)$$

$$\text{for } i \in \{1, 2, \dots, N\}:$$

$$\text{idx} = \arg \text{topk}(\hat{A}_k[i, :])$$

$$\hat{A}_k[i, j] = \begin{cases} 1 & \text{if } j \in \text{idx} \\ 0 & \text{otherwise} \end{cases}.$$

The learnable weight matrices $\mathbf{W}_1 \in \mathbb{R}^{F \times d}$ and $\mathbf{W}_2 \in \mathbb{R}^{F \times d}$ project the feature space into a latent dimension $d$. The use of tanh activation ensures bounded transformations, while the SoftMax operation normalizes the learned adjacency matrix, making it suitable for graph convolution. Although the subsequent top-$k$ operation determines the final sparse graph structure, the softmax normalization plays an important role during training by constraining the adjacency scores to a comparable scale, preventing uncontrolled growth of bilinear similarities, and ensuring stable gradient propagation when jointly optimizing the adjacency learning module with downstream prediction and regularization objectives. The function $\arg \text{topk}(\cdot)$ retrieves the indices of the top-$k$ largest values within a vector, ensuring that only the most relevant connections are retained in the learned graph structure.

The dynamic graph learning equation is reasonably designed. First, the dynamic nature of extreme weather events causes spatial relationships to evolve, making a static adjacency matrix insufficient. By constructing

$\hat{A}_k$ dynamically for each event, the model can capture how weather features, like wind speed and soil moisture, influence spatial dependencies specific to that event. For instance, during a storm, geographic locations in the path of high wind speeds may form stronger dependencies, which are dynamically reflected in $\hat{A}_k$. Additionally, the bilinear operation $\tanh(X_k \mathbf{W}_1) \cdot \tanh(\mathbf{W}_2^\top X_k^\top)$ encodes pairwise interactions between node features. This approach is effective in datasets where features vary significantly across events and locations, allowing the model to identify how one location's features influence another.

To guide the learning of dynamic adjacency matrices, we incorporate external structural priors during training. The regularization is applied to the continuous adjacency scores produced before top-$k$ sparsification, allowing the prior to directly shape the learned affinities through gradient-based optimization. Unlike static geographic relationships, these priors reflect how spatial dependencies evolve under varying weather conditions. Encouraging alignment between the inferred graph structure and event-specific priors helps mitigate the risk of learning spurious connections and improves the model's adaptability to diverse extreme weather scenarios. To achieve this, we impose a regularization loss that minimizes the mean squared error between the learned adjacency matrix $\hat{A}_k$ and the prior adjacency matrix $A_k$:

$$\mathcal{L}_{\text{adj}} = \frac{1}{N^2} \sum_{i=1}^{N} \sum_{j=1}^{N} \left\| A_k[i,j] - \hat{A}_k[i,j] \right\|^2. \tag{2}$$

Furthermore, the integration of this module into the graph convolutional network allows for end-to-end learning. The parameters $\mathbf{W}_1$ and $\mathbf{W}_2$ are jointly trained with the model's other components, ensuring that the learned adjacency matrix aligns with the outage prediction task. This alignment is critical in maximizing the utility of the graph structure for capturing spatial dependencies that directly impact outages.

## 4.2 Hybrid Graph Convolutional Module

We adopt a multi-layer Graph Convolutional Network (Kipf & Welling, 2016) with residual connections as our spatial embedding block to effectively aggregate neighborhood information. GCNs capture complex relational dependencies between nodes by leveraging message passing and neighborhood aggregation, following the layer-wise propagation rule:

$$\mathbf{H}^{(l+1)} = \sigma \left( \tilde{\mathbf{D}}^{-1/2} \tilde{\mathbf{A}} \tilde{\mathbf{D}}^{-1/2} \mathbf{H}^{(l)} \mathbf{W}^{(l)} \right), \tag{3}$$

where $\tilde{\mathbf{A}} = \mathbf{A} + \mathbf{I}_N$ is the adjacency matrix with added self-connections, $\tilde{\mathbf{D}}_{ii} = \sum_j \tilde{\mathbf{A}}_{ij}$ is the degree matrix, $\mathbf{W}^{(l)}$ is the trainable weight matrix at layer $l$, and $\sigma(\cdot)$ is a non-linear activation function such as ReLU. The initial node representation is $\mathbf{H}^{(0)} = \mathbf{X}$, where $\mathbf{X}$ denotes the input feature matrix.

Next, we design a novel hybrid encoding architecture tailored for extreme weather-induced outage prediction based on the standard GCN spatial embedding block. Unlike traditional graph-based approaches that treat all input features uniformly (Kipf & Welling, 2016; Velickovic et al., 2017), our framework separates the processing of static and dynamic features to capture distinct spatial relationships. Specifically, we employ two parallel GCN embedding channels:

- The **static channel** uses a fixed adjacency matrix across all events, modeling persistent geographic or infrastructural relationships. It takes as input the static feature matrix $\mathbf{X}^{(s)}$.

- The **dynamic channel** constructs an event-specific adjacency matrix $\hat{\mathbf{A}}_k$ to reflect the evolving spatial dependencies under different weather conditions. It processes the dynamic feature matrix $\mathbf{X}_k^{(d)}$ for each event $k$.

The outputs of both channels are concatenated to form a unified node embedding, which is then passed through a two-layer MLP to generate the final prediction. This hybrid structure enables more context-aware message passing and improves model performance in scenarios characterized by heterogeneous and evolving spatial dependencies.

### 4.3 Contrastive Learning Across Intra and Inter-Event

Accurate power outage prediction during extreme weather events further requires modeling robust and generalizable spatial representations that capture both local dependencies and global variations across events (Sun et al., 2020; Hassani & Khasahmadi, 2020). However, due to the inherent imbalance in outage datasets, the learned representations may become biased toward dominant outage patterns, limiting the model's ability to generalize. To consider this, we develop a contrastive learning module that enhances the spatial representations learned by the hybrid GCN module to further distinguish similar and dissimilar locations within individual events and across different events (Zhou et al., 2020; Zhang et al., 2023; Velickovic et al., 2019; Zhang et al., 2022b; Xie et al., 2022). By leveraging both intra-event and inter-event contrastive learning strategies, this module improves the quality of the learned representations and ensures the model's ability to generalize across diverse weather conditions. To capture meaningful spatial dependencies within and across different events, we adopt a contrastive learning framework that integrates intra-event and inter-event objectives.

**Intra-event contrast.** Within each event $k$, the learned dynamic adjacency matrix $\hat{A}_k$ defines the spatial relationships among different locations. We treat node pairs connected by an edge in $\hat{A}_k$ as positive pairs, reflecting strong local dependencies under specific weather conditions. In contrast, unconnected node pairs within the same event are considered negative pairs. Let $z_{k,i}$ denote the embedding of node $i$ in event $k$. The intra-event contrastive objective encourages embeddings of positive pairs $(z_{k,i}, z_{k,j})$ to be closer in the representation space, while pushing negative pairs apart. This promotes spatial coherence and localized discriminative learning.

**Inter-event contrast.** Extreme weather events often exhibit distinct spatial patterns. To improve generalization across events, we introduce an inter-event contrastive objective that explicitly contrasts embeddings of nodes from different events. Specifically, for each node $z_{k,i}$, we randomly sample a node $z_{k',m}$ from another event $k' \neq k$ to form an inter-event negative pair. This encourages the model to distinguish between structurally different events and avoid overfitting to event-specific noise.

**Overall contrastive objective.** The final contrastive loss integrates both intra-event and inter-event components and is defined as:

$$\mathcal{L}_c = -\mathbb{E}\left[\log \frac{\exp\left(\text{sim}(z_{k,i}, z_{k,j})/\tau\right)}{\sum_{l \in \mathcal{P}^i \cup \mathcal{N}^i_{\text{intra}} \cup \mathcal{N}^i_{\text{inter}}} \exp\left(\text{sim}(z_{k,i}, z_l)/\tau\right)}\right] \tag{4}$$

where $\text{sim}(\cdot, \cdot)$ denotes cosine similarity between two embeddings; $\mathcal{P}^i$ is the set of positive indices for anchor node $z_{k,i}$; $\mathcal{N}^i_{\text{intra}}$ and $\mathcal{N}^i_{\text{inter}}$ denote intra-event and inter-event negative sets, respectively, and $\tau$ is a temperature scaling parameter.

By optimizing this contrastive loss, the model learns spatially aware and event-generalizable representations, ultimately improving the robustness and accuracy of outage forecasting across diverse extreme weather scenarios.

### 4.4 Optimization Objectives

To effectively optimize the designed model, we adopt the Huber loss (Barron, 2019) between predicted outage counts $\hat{Y}_{k,i}$ and ground truths $Y_{k,i}$ as the main learning objective $\mathcal{L}_p$. The Huber loss is defined as:

$$\mathcal{L}_p(y, \hat{y}) = \begin{cases} \frac{1}{2}(Y_{k,i} - \hat{Y}_{k,i})^2, & \text{if } |Y_{k,i} - \hat{Y}_{k,i}| \leq \delta \\ \delta \cdot (|Y_{k,i} - \hat{Y}_{k,i}| - \frac{1}{2}\delta), & \text{otherwise} \end{cases} \tag{5}$$

where $\delta$ is a threshold parameter that balances the sensitivity to outliers. The Huber loss allows us to effectively handle predictions across a wide range of weather scenarios. The total learning objective function for this regression task consists of the forecasting objective $\mathcal{L}_p$, the contrastive learning objective $\mathcal{L}_c$, and the learning dynamic adjacency matrix objective $\mathcal{L}_{adj}$:

$$\mathcal{L}_{total} = \mathcal{L}_p + \lambda\mathcal{L}_c + \gamma\mathcal{L}_{adj}, \tag{6}$$

where $\lambda \geq 0$ and $\gamma \geq 0$ are hyperparameters determined by grid search over the training set.

## 5 Experiment

### 5.1 Experiment Setup

#### 5.1.1 Datasets and Preprocessing

Table 1: The detailed statistics of four datasets

| Datasets | # Events | # Locations | Feature Length | Output Length |
|---|---|---|---|---|
| Connecticut | 294 | 815 | 390 | 1 |
| New Hampshire | 227 | 1022 | 390 | 1 |
| Western Massachusetts | 271 | 312 | 390 | 1 |
| Eastern Massachusetts | 231 | 383 | 390 | 1 |

To evaluate the effectiveness of the proposed SA-HGNN to predict power outages during extreme weather events, we compare SA-HGNN with baseline methods over four utility service territories of Eversource, *i.e.*, Connecticut, Western Massachusetts, Eastern Massachusetts, and New Hampshire. The detailed statistics of benchmark datasets are summarized in Table 1.

The datasets include weather variables, utility infrastructure, land cover, vegetation, and historical outage data for each storm, modeling power disruptions across uniformly distributed locations within each state. Details on feature sources are provided below.

**Weather**: Weather input is the largest source of uncertainty in the outage predictions (Guikema, 2018). The weather data is obtained from 48-hour simulations using version 3.8.1 of the Advanced Weather Research and Forecasting (WRF) model (Powers et al., 2017; Skamarock et al., 2008), with a 4-km horizontal grid spacing over the northeastern United States. Initial and lateral conditions were derived from the North American Mesoscale (NAM) Forecast System at a 12-km spatial resolution.

**Utility Infrastructure**: The utility infrastructure information of Eversource Energy contains multiple types of assets, including electric fuses, reclosers, and poles. These serve as key explanatory variables because outages are recorded at the asset level, and the risk of having a reported outage is directly proportional to the number of assets.

**Land Cover**: The U.S. Geological Survey (USGS) provided National Land Cover Database (NLCD) products, detailing vegetation and urbanization patterns. Since tree interaction with overhead lines during storms is a major cause of outages, we incorporated tree-related land cover variables, including the percentages of miscellaneous forests, deciduous forests, and developed areas.

The complete extreme weather database contains 390 distinct numerical features associated with outage occurrences. Incorporating all features in a high-dimensional space introduces challenges that degrade model performance and interpretability. The curse of dimensionality leads to data sparsity, poor generalization, and overfitting, while redundant or irrelevant features add noise, increase computational costs, and complicate model training (Rice et al., 2020; Bejani & Ghatee, 2021). To address these issues and enhance model efficiency, we applied the Pearson correlation coefficient to quantify the linear relationship between each feature $X_m$ and the outage count $Y$. The Pearson correlation coefficient $r_{X_m, Y}$ is defined as:

$$r_{X_m, Y} = \frac{\sum_{i=1}^{N} \left( X_{m,i} - \bar{X}_m \right) \left( Y_i - \bar{Y} \right)}{\sqrt{\sum_{i=1}^{N} \left( X_{m,i} - \bar{X}_m \right)^2} \sqrt{\sum_{i=1}^{N} \left( Y_i - \bar{Y} \right)^2}}, \tag{7}$$

where $X_{m,i}$ is the value of feature $X_m$ for location $i$, $\bar{X}_m$ is the mean of feature $X_m$ across events, $Y_i$ is the outage count for location $i$, and $\bar{Y}$ is the mean outage count. Based on the computed correlation values, we selected the top 50 features with the highest correlation to outage counts. Among the selected variables, 20 features are static, representing location-specific characteristics that remain static over time, while the remaining 30 features are dynamic, capturing temporal variations across different events. The

complete selected feature description is provided in Appendix A. We construct two adjacency matrices based on these two types of features and use them as external structural knowledge. The static adjacency matrix is derived from geographic distances, where each location is connected to its eight nearest neighbors, forming a fixed graph structure that remains consistent across all events. In contrast, the dynamic adjacency matrix captures event-specific spatial relationships that may vary under different weather conditions. To construct this matrix, we compute the Pearson correlation coefficients between dynamic features measured during each event. For a given location, we identify the eight locations with the highest Pearson correlation coefficients and establish eight edges between them. As a result, each event in our dataset is associated with a unique dynamic adjacency matrix, which serves as external structural knowledge to guide the dynamic graph learning module in Section 4.1.

### 5.1.2 Evaluation Protocols and Metrics

Given the critical importance of every extreme weather event, we adopt a leave-one-out evaluation strategy to assess model performance (Yang et al., 2021). For example, in the Connecticut dataset, which consists of 294 extreme weather events, we conduct 294 training and evaluation loops, each time leaving out a single event for evaluation while training on the remaining 293 events. A similar leave-one-out procedure is applied to datasets from Western Massachusetts, Eastern Massachusetts, and New Hampshire.

We evaluated the performance of outage predictions by using absolute error ($AE$), absolute percentage error ($APE$), mean absolute percentage error ($MAPE$), centered root mean square error ($CRMSE$), and R-square ($r^2$). The definitions of evaluation metrics are detailed in Appendix B.

### 5.1.3 Baseline Model

To establish a comprehensive baseline for our proposed method, we conducted a thorough comparison against 7 models, which we have categorized into three groups: traditional machine learning methods: Random Forest (Breiman, 2001), XGBoost (Chen & Guestrin, 2016); GNN-based methods: ChebNet (Tang et al., 2024), Graph Attention Networks (GAT) (Velickovic et al., 2017), GraphSAGE (Hamilton et al., 2017b), Graph Isomorphism Network (GIN) (Xu et al., 2018); tabular foundation model: TabPFN (Hollmann et al., 2025). We provide detailed descriptions of each baseline below:

- **Random Forest** (Breiman, 2001): This method combines multiple decision trees using bagging to improve predictive performance and reduce overfitting, making it robust for handling high-dimensional data and capturing complex relationships.

- **XGBoost** (Chen & Guestrin, 2016): This is a gradient boosting framework that builds decision trees sequentially, optimizing for speed and accuracy.

- **Graph Attention Networks (GAT)** (Velickovic et al., 2017): It leverages attention mechanisms to dynamically assign weights to neighboring nodes, enabling the model to focus on the most relevant parts of the graph for learning node embeddings.

- **GraphSAGE** (Hamilton et al., 2017b): The method generates node embeddings by sampling and aggregating features from a fixed-size neighborhood, enabling efficient and scalable learning on large graphs.

- **Graph Isomorphism Network (GIN)** (Xu et al., 2018): GIN achieves strong expressive power by using a sum aggregation function and learnable weights, making it capable of distinguishing different graph structures more effectively than traditional GNNs.

- **TabPFN** (Hollmann et al., 2025): It is a pre-trained transformer-based deep learning foundation model for regression and classification on tabular data.

Table 2: Extreme weather outage prediction results. Best results are highlighted in **bold**, and the second best results are underlined.

| Datasets | Metric | SA-HGNN | Random Forest | XGBoost | GAT | GIN | GraphSAGE | TabPFN |
|---|---|---|---|---|---|---|---|---|
| **CT** | AE q25 | **25.00** | 84.50 | 48.00 | 31.50 | 33.00 | 32.00 | 31.25 |
| | AE q50 | **62.50** | 199.00 | 136.00 | 78.00 | 63.50 | 77.50 | 74.00 |
| | APE q25 | **23.09** | 45.75 | 32.10 | 29.65 | 24.73 | 28.57 | 24.97 |
| | APE q50 | **49.36** | 109.62 | 64.61 | 55.32 | 52.72 | 54.69 | 52.48 |
| | MAPE | **52.77** | 127.15 | 155.87 | 65.52 | 65.98 | 67.87 | 79.41 |
| | CRMSE | **851** | 1726 | 1323 | 1598 | 1755 | 1761 | 1590 |
| | $R^2$ | **0.79** | 0.23 | 0.49 | 0.25 | 0.10 | 0.12 | 0.26 |
| **NH** | AE q25 | 12.50 | 37.00 | 26.00 | 15.00 | **12.00** | 13.00 | 13.00 |
| | AE q50 | 29.00 | 83.00 | 58.00 | **27.00** | 32.00 | 31.00 | 31.00 |
| | APE q25 | 23.75 | 41.42 | 28.34 | **20.14** | 21.29 | 22.16 | 23.40 |
| | APE q50 | **40.70** | 88.94 | 70.24 | 41.30 | 46.75 | 42.31 | 48.45 |
| | MAPE | **45.66** | 190.77 | 142.54 | 52.34 | 51.38 | 59.89 | 51.51 |
| | CRMSE | 364 | **359** | 360 | 388 | 389 | 379 | 395 |
| | $R^2$ | 0.09 | **0.11** | **0.11** | 0.04 | 0.04 | 0.01 | 0.07 |
| **WMA** | AE q25 | 5.00 | 10.00 | 8.00 | **4.00** | 4.50 | **4.00** | 5.00 |
| | AE q50 | **9.00** | 26.00 | 19.00 | 9.00 | 10.00 | 9.00 | 10.00 |
| | APE q25 | 25.54 | 32.44 | 31.15 | **25.00** | 26.79 | 25.89 | 25.90 |
| | APE q50 | 48.65 | 90.91 | 69.23 | 47.06 | 50.00 | **46.67** | 48.28 |
| | MAPE | 55.87 | 177.64 | 143.29 | 55.76 | 54.52 | **51.81** | 54.98 |
| | CRMSE | **92** | 116 | 110 | 111 | 116 | 116 | 112 |
| | $R^2$ | **0.32** | 0.09 | 0.02 | 0.10 | 0.08 | 0.08 | 0.02 |
| **EMA** | AE q25 | **11.00** | 28.00 | 21.00 | 12.00 | 11.00 | 12.00 | 12.50 |
| | AE q50 | **21.00** | 55.00 | 44.00 | 25.00 | 25.00 | 25.00 | 23.00 |
| | APE q25 | 22.02 | 39.91 | 28.87 | 23.75 | 21.21 | **20.24** | 22.78 |
| | APE q50 | **39.02** | 91.76 | 66.67 | 43.96 | 41.67 | 45.16 | 40.91 |
| | MAPE | **50.19** | 170.57 | 144.31 | 60.09 | 56.61 | 64.14 | 54.33 |
| | CRMSE | **433** | 566 | 564.26 | 550 | 575 | 547 | 522 |
| | $R^2$ | **0.43** | 0.03 | 0.04 | 0.09 | 0.10 | 0.09 | 0.17 |

## 5.2 Main Results

Table 2 shows the full outage prediction results on all four datasets: Connecticut, New Hampshire, Western Massachusetts, and Eastern Massachusetts. While SA-HGNN consistently achieves competitive or superior performance across all four datasets, the magnitude of improvement varies across regions. In particular, the most significant gains are observed in Connecticut, whereas improvements in other regions are more moderate. This variation is closely related to differences in spatial dependency structures across regions, which directly affect the extent to which spatial modeling can improve prediction performance. In particular, Connecticut exhibits stronger and more coherent inter-location dependencies during extreme weather events, allowing SA-HGNN to effectively leverage dynamic graph structures. Thus the improvement of SA-HGNN in terms of MAPE is the most significant, improving by 19.45% compared to the second best method. Meanwhile, it achieves the best (lowest) AE Q50 and APE Q50, showcasing a clear advantage over three other baseline models in Figure 2. In contrast, other regions tend to exhibit more localized or fragmented outage patterns, which reduce the benefit of modeling long-range spatial interactions. Importantly, even in these cases, SA-HGNN remains competitive with existing approaches and does not degrade performance, demonstrating robustness across diverse regional characteristics. SA-HGNN outperforms ensemble learning methods methods such as Random Forest and XGBoost, graph-based models including GAT, GIN, and GraphSAGE, as well as the state-of-the-art tabular foundation model TabPFN. The model achieves the

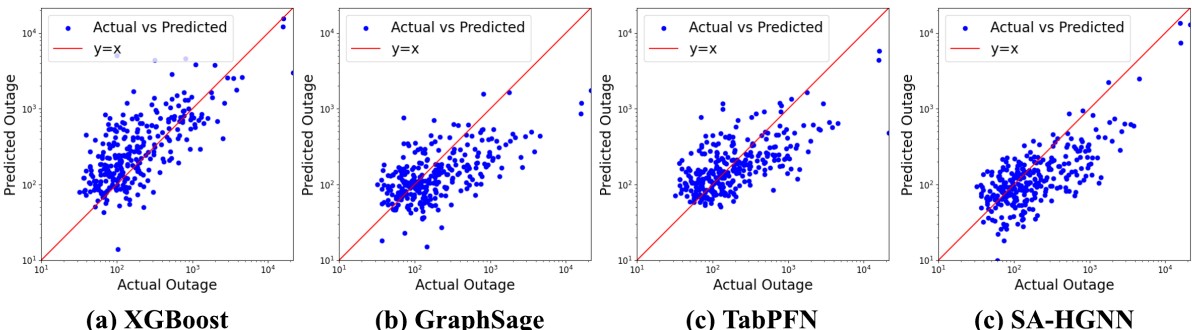

Figure 2: Actual outages vs. predicted outages comparison of four models on Connecticut extreme weather data.

highest $R^2$ scores in multiple regions, reaching 0.79 in Connecticut and 0.43 in Eastern Massachusetts, indicating superior predictive accuracy. Additionally, SA-HGNN demonstrates the lowest CRSME values, showing improved robustness in capturing outage patterns. These results confirm that SA-HGNN effectively models spatial dependencies and dynamic weather impacts, leading to more accurate outage forecasts.

Our experimental results show that ensemble learning methods, such as Random Forest and XGBoost, yield suboptimal performance in outage prediction. This is primarily because they treat each location independently, ignoring critical spatial correlations and dynamic interactions that shape outage patterns during extreme weather events. In contrast, graph-based models like GAT and GIN generally perform better by leveraging graph structures to model spatial dependencies. However, their aggregation mechanisms are often fixed, and their adjacency matrices are typically predefined, limiting their adaptability to event-specific spatial dynamics. For instance, while GAT assigns attention-based weights to neighboring nodes, it does not dynamically adjust the graph structure to reflect evolving event-specific relationships.

SA-HGNN improves outage prediction by effectively capturing the unique spatial dependencies and event-specific dynamics of extreme weather data. Unlike traditional models, SA-HGNN processes static and dynamic features separately, ensuring better representation of both constant infrastructure-related attributes and evolving weather conditions, as demonstrated in Section 5.3. A key advantage of SA-HGNN is its dynamic adjacency matrix, which adapts to each event and captures event-specific spatial relationships crucial for predicting outages under varying weather conditions. Additionally, the contrastive learning module enhances the model's ability to distinguish intra-event neighbors from inter-event non-neighbors, leading to more robust and generalizable node embeddings. This is evident in the t-SNE visualization, where embeddings with similar outage counts form well-defined clusters after contrastive learning, highlighting its effectiveness in learning meaningful representations as shown in Figure 3. And we observe that regions with stronger inter-location correlations tend to benefit more from spatially adaptive modeling, which is consistent with the design of SA-HGNN and explains the variation in performance gains across datasets.

### 5.3 Ablation Study

In this section, we conduct an ablation study on the Connecticut extreme weather dataset to assess the impact of key components on model performance as shown in Table 3. Our analysis highlights three critical modules that contribute to SA-HGNN's effectiveness:

**w/o HGNN:** SA-HGNN without the hybrid graph convolution module, which separates the processing of dynamic and constant neighbor information into two distinct branches. In this configuration, the module is replaced with a single graph convolution branch that exclusively considers static neighbor information. With this setting, the performance consistently perform worse than SA-HGNN. This is because during extreme

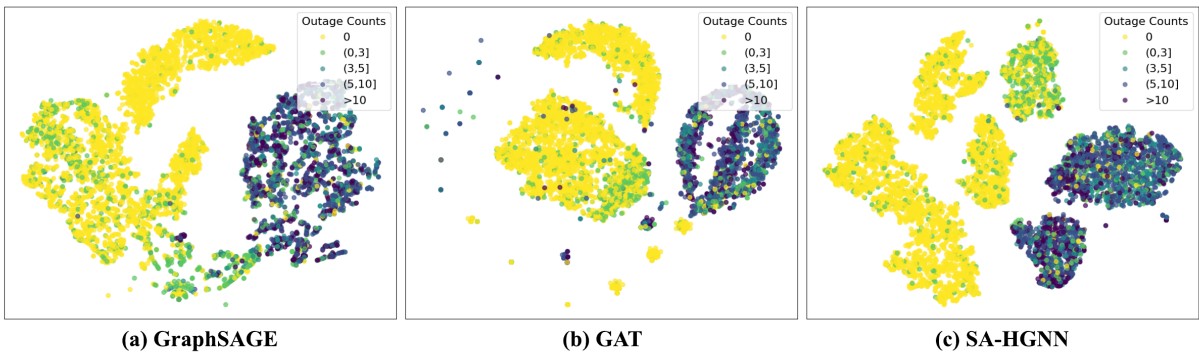

(a) GraphSAGE         (b) GAT         (c) SA-HGNN

Figure 3: Comparison of learned location embeddings across different methods on the Connecticut dataset.

weather events, the outage of one location only only rely on its geographical neighborhood locations but also depend on the dynamic weather conditions across different locations.

**w/o DSK:** SA-HGNN without dynamic structural knowledge (DSK) guiding the learning process of the dynamic adjacency matrix. Instead, the model derives the dynamic adjacency matrix directly from each weather event's data, without incorporating external prior information. The absence of external dynamic structural knowledge for each weather event restricts the model's ability to effectively capture the complex and evolving spatial dependencies that arise under different extreme weather conditions. Without dynamic graph learning module, the graph learning process struggles to generalize to new weather events. Consequently, the learned dynamic graphs are less informative, leading to suboptimal performance.

**w/o CL:** SA-HGNN without contrastive learning, where the SA-HGNN is trained to generate location embeddings without explicitly grouping similar location pairs or enforcing discrimination between different types of locations. The results show that removing contrastive learning degrades model performance, underscoring its role in enhancing outage prediction. In addition, we visualized the embeddings of the learned locations in eight extreme weather events using t-SNE, as shown in Figure 4, where each event contains 815 locations. After applying contrastive learning, the embeddings exhibit clear separability and are grouped with similar outage counts. However, the absence of contrastive

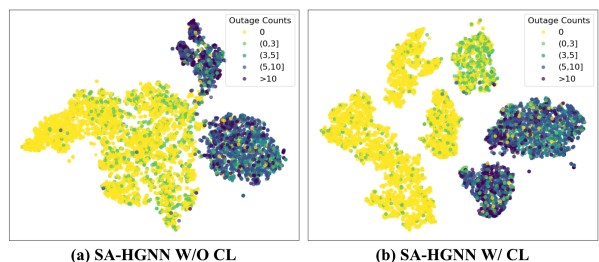

(a) SA-HGNN W/O CL      (b) SA-HGNN W/ CL

Figure 4: SA-HGNN CL representation comparison

learning significantly degrades the embedding structure, resulting in less distinguishable representations. By minimizing intra-event distances between similar locations while maximizing inter-event distances across all locations, contrastive learning can help overcome potential imbalance issues and improve the model's capability to capture meaningful spatial patterns.

### 5.4 Case Study

We conduct case studies on two representative extreme weather events to further demonstrate the effectiveness of the proposed SA-HGNN model, as illustrated in Figures 5 and 6. Figure 5 visualizes the predicted outage distribution across the Connecticut service territory during Hurricane Irene (August 28, 2011). While GAT partially captures spatial outage patterns, GIN fails to accurately predict high-outage locations, and XGBoost tends to overestimate outage counts in low-impact areas, resulting in notable deviations from the observed outage distribution. The outage prediction patterns indicate that SA-HGNN effectively captures local outage patterns and aligns most closely with the ground truth distribution in Figure 5(a), highlighting the advantages of the proposed SA-HGNN.

Table 3: Ablation study of our proposed SA-HGNN.

| Methods | SA-HGNN | w/o HGNN | w/o DSK | w/o CL |
|---------|---------|----------|---------|--------|
| AE Q25  | **25.00** | 30.50 | 31.00 | 34.00 |
| AE Q50  | **62.50** | 68.00 | 67.50 | 77.50 |
| APE Q25 | **23.09** | 30.18 | 30.53 | 36.92 |
| APE Q50 | **49.36** | 55.18 | 49.85 | 57.46 |
| MAPE    | 52.77 | 53.46 | **52.11** | 56.80 |
| CRMSE   | **851** | 1565 | 1405 | 1378 |
| $R^2$   | **0.79** | 0.29 | 0.42 | 0.45 |

Figure 6 presents a second case study for a severe storm event in Eastern Massachusetts (October 29, 2017). Compared to Connecticut, this event exhibits sparser and more fragmented outage patterns. Despite the weaker spatial signal, SA-HGNN remains more consistent with the ground truth distribution than the baselines, demonstrating robustness across regions with differing outage characteristics.

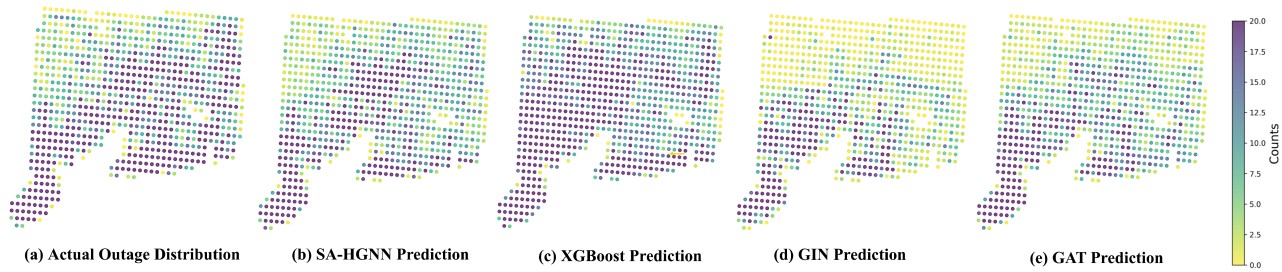

(a) Actual Outage Distribution  (b) SA-HGNN Prediction  (c) XGBoost Prediction  (d) GIN Prediction  (e) GAT Prediction

Figure 5: Outage prediction distribution of Hurricane Irene (August 28, 2011) in Connecticut. The ground truth total outage count is 16022 (a). SA-HGNN (b) produces the closest total prediction (14512) compared to XGBoost (21385), GIN (11273), and GAT (13742).

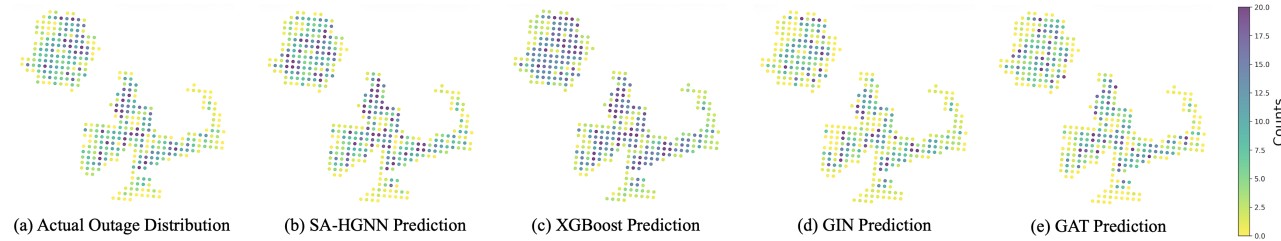

(a) Actual Outage Distribution  (b) SA-HGNN Prediction  (c) XGBoost Prediction  (d) GIN Prediction  (e) GAT Prediction

Figure 6: Outage prediction distribution of Storm (October 29, 2017) in East Massachusetts. The ground truth total outage count is 2520 (a). SA-HGNN (b) produces the closest total prediction (2673) compared to XGBoost (3052), GIN (2087), and GAT (2201).

## 5.5 Hyperparameter Sensitivity

We further examine the sensitivity of SA-HGNN to the weighting coefficients $\lambda$ and $\gamma$ in Eq. (6), which control the relative contributions of the contrastive objective and the dynamic adjacency regularization, respectively. As shown in Figure 6, the model demonstrates stable performance over a wide range of values for both hyperparameters. In particular, moderate values of $\lambda$ lead to consistently low MAPE, AE$q25$, and APE$q25$, indicating that the contrastive objective enhances representation learning without overwhelming the forecasting loss. The performance is also relatively insensitive to $\gamma$, and based on this analysis, we set $\lambda = 0.01$ and $\gamma = 0.5$ in the experiments, which achieve a good balance between accuracy and stability. Overall, these results show that the proposed framework is robust and does not rely on fine-grained hyperparameter tuning.

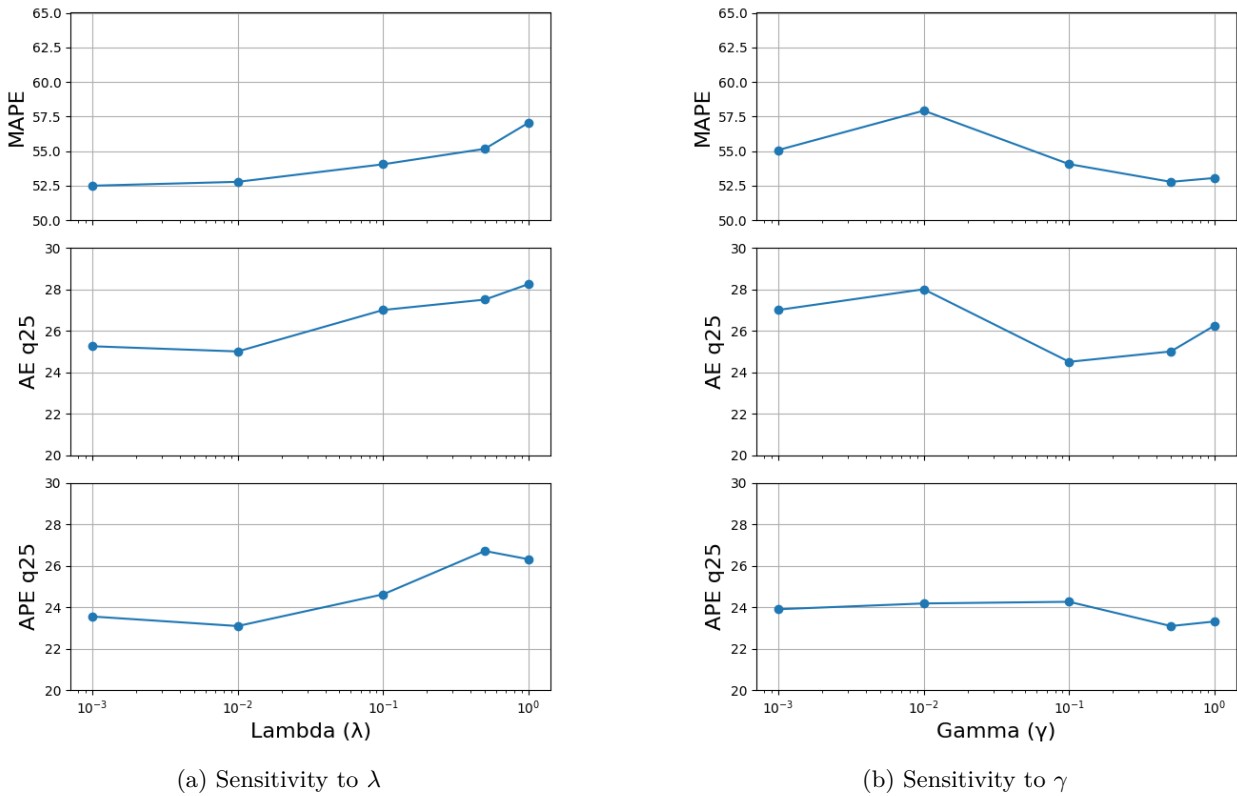

(a) Sensitivity to $\lambda$           (b) Sensitivity to $\gamma$

Figure 7: Hyperparameter sensitivity analysis. We report MAPE, AE q25, and APE q25 under different values of $\lambda$ and $\gamma$. The x-axis is shown in log scale.

## 6 Limitations and Future Work

Although our proposed framework SA-HGNN demonstrates promising performance in modeling both static infrastructure features and dynamic storm-level spatial dependence, several limitations remain that motivate future research.

One limitation of the current framework is that data from different service territories are treated independently, with separate models trained for each region. Although the data are collected from multiple real-world service territories, these regions exhibit inherent distributional differences in infrastructure, vegetation, and climatic conditions, leading to feature drift across territories. Training region-specific models avoids information leakage and allows the model to capture local characteristics, but it does not explicitly leverage shared structure across regions, nor does it enable a systematic evaluation of cross-region generalization. As a result, the model's portability to utilities or service territories with substantially different characteristics has not yet been studied. Addressing such cross-region heterogeneity is an important direction for future work, and recent advances in domain generalization and distribution shift modeling (Zhou et al., 2022; Long et al., 2023) could be explored to improve robustness and generalizability across heterogeneous service territories and utility systems.

Another challenge arises from the temporal granularity of the available data. Each extreme weather event in our dataset is represented by a single aggregated snapshot of meteorological and infrastructure variables, rather than a temporally continuous sequence that tracks storm evolution. As a result, the constructed dynamic adjacency matrix varies across events but remains static within each event, limiting the model's ability to capture the evolving propagation of storms and cascading outage dynamics over time. Recent work on spatio-temporal graph learning emphasizes the importance of jointly modeling spatial and temporal dependencies through dynamic edge adaptation (Corradini et al., 2025). In addition, recent surveys on

multi-modal time series analysis highlight the importance of temporally resolved and multi-source signals for modeling complex real-world systems (Jiang et al., 2025). Incorporating temporally resolved observations, such as hourly wind fields or sequential outage reports, would enable future extensions toward fully spatio-temporal graph formulations, where both node states and edge connections evolve continuously to reflect the real-time progression of extreme weather systems.

Finally, the contrastive learning strategy improves representation discriminability, but the selection of positive and negative pairs is heuristic and may not fully reflect complex inter-event dependencies. Recent advances in self-supervised and contrastive learning for structured data have explored more adaptive sampling strategies and robust objectives (Jiang et al., 2024). Future research could explore adaptive contrastive sampling or curriculum-based contrastive learning to better capture hierarchical relationships among events of varying severity.

## 7 Conclusion

In this study, we introduced SA-HGNN, a spatially aware hybrid graph neural network to predicter outages caused by extreme weather events. By integrating dynamic graph inference, hybrid graph convolution, and contrastive learning, SA-HGNN effectively captures both static and evolving spatial dependencies. Experimental results on four utility service territories show that SA-HGNN outperforms existing ensemble and graph-based models by adapting to event-specific graph structures and refining node embeddings.

Beyond outage prediction, this research contributes to the broader field of graph-based forecasting under dynamic conditions, with potential applications in disaster response, climate impact modeling, and resilient infrastructure planning.

## Acknowledgements

This work was supported in part by the National Science Foundation under Grant No. 2338878, the NSF WISER IUCRC, and the UConn Eversource Energy Center.

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

# A    Selected Feature Information

This section provides a comprehensive list of all the selected features along with their explanations. Each feature is described in detail to clarify its significance and relevance to the analysis. These features encompass various environmental, meteorological, and infrastructure-related factors essential for understanding outage patterns.

**land21**: Land cover Area - Developed, Open Space
**land22**: Land cover Area - Developed, Low Intensity
**land23**: Land cover Area - Developed, Medium Intensity
**land24**: Land cover Area - Developed High Intensity
**land43**: Land cover Area - Mixed Forest
**landTotal**: Land cover Area - Total
**prec81**: Percentage of land81(Land cover - Pasture/Hay)
**soilDepth**: Mean Soil Depth
**avgSMOIS3**: Average Soil Moisture (40-100 cm deep)
**stdSMOIS4**: Standard Deviation Soil Moisture (100-200 cm deep)
**avgTPA**: Trees per acre **avgLFSH**: Leaf Stress
**avgCIN**: Average Convective Inhibition
**stdCIN**: Standard Deviation Convective Inhibition
**avgDPT**: Avearage Dew Point Temperature at 2 m
**stdDPT**: Standard Deviation Dew Point Temperature at 2 m
**hydNo**: Percent Not Hydric Soils
**avgSDI**: Average Total stand density index
**avgHardBA**: Average Hardwood basal area
**stdHardBA**: Standard Deviation Hardwood basal area
**avgHardSDI**: Average Hardwood stand density index
**stdHardSDI**: Standard Deviation Hardwood stand density index
**peakPSFC**: Peak Surface Pressure
**minPSFC**: Minimum Surface Pressure
**peakPOTT**: Peak Potential Temperature at 2 m

**stdPOTT**: Standard Deviation Potential Temperature at 2 m
**peakSPFH**: Peak Specific Humidity
**HGT**: WRF elevation
**coggt17**: Duration of Continuous Gusts above 17 m/s
**coggt27**: Duration of Continuous Gusts above 27 m/s
**coggt22**: Duration of Continuous Gusts above 22 m/s
**ggt17**: Hours of Wind Gusts above 17 m/s
**ggt22**: Hours of Wind Gusts above 22 m/s
**ggt27**: Hours of Wind Gusts above 27 m/s
**peakGUST**: Peak Wind Gust Speed
**maxGUST**: Max Wind Gust Speed
**stdGUST**: Standard Deviation Wind Gust Speed
**peakW850**: Peak Wind Speed above 850 mb
**maxW850**: Max Wind Speed above 850 mb
**stdW850**: Standard Deviation Wind Speed at 850 mb
**stdTDIF**: Standard Deviation of Temperature Difference (850 mb to 1000 mb)
**maxWSPD**: max Wind Speed at 10m
**peakLLWS**: Low Level Wind Shear
**avgCAPE**: Convective Available Potential Energy
**maxCAPE**: Max Convective Available Potential Energy
**stdTURB**: Standard Deviation Turbulence
**maxTURB**: Max Turbulence
**poleCount**: Number of poles
**fuseCount**: Number of fuses
**ohLength**: Length of overhead lines
**reclrCount**: Number of reclosers

## B   Evaluations

AE is used to measure the difference between the total predicted $(p_i)$ and actual $(o_i)$ outage counts from event $i$. AE Q25 and AE Q50 mean the 25th and 50th quantile value of all events' AE. And AE is calculated as:

$$AE = |p_i - o_i|, \tag{8}$$

Also APE Q25 and APE Q50 represents the 25th and 50th quantile value of all events' APE, which is calculated as:

$$APE = \frac{|p_i - o_i|}{o_i}, \tag{9}$$

MAPE is utilized to mean relative error as a percentage and is defined as:

$$\text{MAPE} = \frac{100\%}{n} \sum_{i=1}^{n} \left| \frac{o_i - p_i}{o_i} \right|, \tag{10}$$

CRMSE is to measure the deviation of predictions from the actual values while removing systematic biases. The lower this value, the better the performance of the model.

$$\text{CRMSE} = \sqrt{\frac{1}{n} \sum_{i=1}^{n} \left( p_i - o_i - \frac{\sum_{i=1}^{n} (p_i - o_i)}{n} \right)^2}, \tag{11}$$

$R^2$ shows the goodness of fit of the various model predictions to the actual outages. The higher this value, the better the performance of the model.

$$R^2 = \frac{1}{n} \sum_{i=1}^{n} \frac{\left( o_i - \frac{\sum_{i=1}^{n} o_i}{n} \right) \left( p_i - \frac{\sum_{i=1}^{n} p_i}{n} \right)}{o_i p_i}, \tag{12}$$

