# OpenReview forum: "Empowering Power Outage Prediction with Spatially Aware Hybrid Graph Neural Networks and Contrastive Learning"
_TMLR — Accepted by TMLR_

### Review · Reviewer_ExvQ · 2025-12-09

**Summary Of Contributions:**

The paper studies the problem of predicting power outages caused by extreme weather events using graph neural networks (GNNs). The issue is to incorporate the spatial effects of these weather events and the imbalance between frequent low-impact events and rare high-impact events.

The paper proposes the Spatially Aware Hybrid Graph Neural Networks (SA-HGNN) to enhance outage predictions. This framework uses a dynamic graph learning module to infer evolving spatial relationships based on event-specific weather features. It also incorporates a contrastive learning module to generate location-specific embeddings, enabling the model to distinguish between similar and dissimilar outage patterns across different events.

Experiments show that SA-HGNN achieves state-of-the-art performance across four utility service territories: Connecticut, Western Massachusetts, Eastern Massachusetts, and New Hampshire. The results demonstrate that the model outperforms baseline methods, including Random Forest, XGBoost, and other graph neural networks.

**Audience:**

Yes

**Audience Explanation:**

This paper contributes to the literature of hybrid GNN architecture and graph contrastive learning. And the application has a real-world impact.

**Broader Impact Concerns:**

N/A.

**Claims And Evidence:**

Yes

**Claims Explanation:**

The main claim is the effectiveness of the proposed method SA-HGNN. This is supported by the empirical validation on four real-world datasets. SA-HGNN outperforms traditional methods (RF, XGBoost), graph-based methods (GAT, GraphSAGE, GIN), and foundation (TabPFN) baselines with multiple metrics (AE, APE, MAPE, CRMSE, $R^2$).

The ablation study shows that each component of SA-HGNN is functional. The ablation “w/o HGNN” (single GCN on static neighbors only) yields much worse $R^2$ (0.29 vs. 0.79) and higher errors on CT, indicating that separating static and dynamic channels is essential. The ablation “w/o DSK” (no dynamic structural prior) also degrades CRMSE and $R^2$, showing that guiding the dynamic graph with event-specific priors is useful. The “w/o CL” ablation consistently worsens AE/APE/MAPE and reduces $R^2$ (0.45 vs. 0.79 on CT), indicating that the contrastive term improves the learned embeddings and final predictions.

**Requested Changes:**

1. In Section 4.1, the authors first use the softmax function to normalize the learned adjacency matrix, and then apply a top-k method to select the indices, after which they convert them into a binary 0/1 encoding. In my understanding, these two operations conflict with each other. If the final result is a 0/1 binary encoding, then softmax does not change the top-k ordering and therefore becomes useless.
2. The authors do not discuss the hyperparameters.
3. In Equation (6), the final loss is a weighted combination of the forecasting objective, the contrastive learning objective, and the learning dynamic adjacency matrix objective. In my understanding, these three objectives are highly heterogeneous. Therefore, a discussion and evaluation of $\lambda$ and $\gamma$ is necessary.

Minor: ChebNet Tang et al. (2024) is mentioned in the subsection 'baseline model', but not reported in Table 3 or any other experiments.

---

> ### Author Response · Authors · 2026-01-09
>
> 1. In Section 4.1, the authors first use the softmax function to normalize the learned adjacency matrix, and then apply a top-k method to select the indices, after which they convert them into a binary 0/1 encoding. In my understanding, these two operations conflict with each other. If the final result is a 0/1 binary encoding, then softmax does not change the top-k ordering and therefore becomes useless.
>
> Thank you for raising this point. We agree that, if viewed purely from the perspective of the final binary adjacency matrix, the softmax normalization does not affect the top-k ordering itself. However, the role of the softmax is not to influence the hard top-k selection at inference time, but to stabilize and regularize the learning process of the adjacency scores during training. Specifically, softmax constrains the pairwise affinity scores to a comparable scale across nodes, prevents uncontrolled growth of bilinear similarities, and ensures well-behaved gradients when optimizing the adjacency learning module jointly with the downstream prediction loss and structural regularization. The subsequent top-k operation enforces sparsity and locality in the inferred graph, which is important for efficient message passing and spatial interpretability. We have revised Section 4.1 to explicitly clarify the distinct roles of softmax (score normalization and training stability) and top-k (sparse graph construction).
>
> 2. The authors do not discuss the hyperparameters.
>
> Thank you for pointing this out. We have added a hyperparameter sensitivity analysis in Section 5.5 (Figure 7), where we systematically study the effects of the weighting coefficients lambda and gamma in Eq. 6. The results show that the proposed SA-HGNN is stable across a wide range of values and does not rely on fine-grained tuning.  We believe this addition addresses the concern and clarifies the role of hyperparameters in the model.
>
> 3. In Equation (6), the final loss is a weighted combination of the forecasting objective, the contrastive learning objective, and the learning dynamic adjacency matrix objective. In my understanding, these three objectives are highly heterogeneous. Therefore, a discussion and evaluation of and is necessary.
>
> Thank you for this comment. In our framework, the forecasting loss provides the primary supervision, while the contrastive objective and the adjacency regularization act as auxiliary terms that shape the learned representations and inferred graph structure. The weighting coefficients lambda and gamma were selected via grid search on the training data and kept fixed across all experiments, and we found the model to be stable within a broad range of values around the selected settings. In the revision, we have explicitly discussed the functional role of each term, and reported the chosen values and search ranges for weights to demonstrate robustness in Section 5.5.

---

### Review · Reviewer_QaSo · 2025-12-15

**Summary Of Contributions:**

This paper is concerned with the prediction of power outages using graph neural networks. The setting is thus: each prediction query is formed by a weather event over there are $N$ locations within a geographic area. Each location offers measurements of $F$ meteorological (or other) features, yielding an instance $X \in \mathbb{R}^{N \times F},$ and the goal is to predict a vector of the number of outages at each location, denoted $Y$. Each event is further modeled by a graph on the $N$ locations, with adjacency matrix $A$, which captures spatial or functional relationships introduced by the weather event. Given that the structural role of the underlying graph, the paper proposes a graph neural network approach to address this task. A significant detail is that the features $X$ are decomposed as static and dynamic features $(X^s, X^d),$ where the former do not change with events, while the latter do.

The paper considers a setting where the training data is composed of triples $(X_k, A_k, Y_k)$, while at test time $Y_k$ must be predicted from the features. The main contribution is a setup for modeling this data where the static and dynamic features are treated together. Specifically, what is effectively a shallow net is constructed for learning a dynamic adjacency matrix on the dynamic data, while the static data is crunched into a static adjacency matrix that is also given as input to the system (following the design proposed in certain recent work). Both are fed into two separate graph convolutional network, and their final representations are concatenated and then passed through an MLP. These representations are further tuned with a softmax contrastive loss thus: 1) within any event, the embeddings of adjacent nodes are treated as positive pairs, and those of non-adjacent nodes as negative pairs. 2) the embeddings of any two nodes across different events are treated as negative pairs. This contrastive loss is added to handle imbalance in outage data (the nature of which is not specified, but presumably outages are fairly rare in, say, new england). Finally, the output $\hat{Y}$ is penalised with a Huber loss, giving a three term (adjacency, contrastive, Huber) loss that is optimized.

Finally, this modeling pipeline is implemented for four outage datasets, and compared to six other methods. Two of these are classical decision forests (random forest, and XGboost), one is a foundation model for tabular data (TabPFN), and three graph neural network architectures. The main observation is that across a number of error metrics, the proposed models are either better than all competitors, or perform similarly to the best competitor.

**Additional Comments:**

I think most citations in the submission should use a \citep command instead of \citet. Compare

page 1, current: "... with the frequency and severity of billion-dollar disasters steadily increasing over the past two decades Campbell (2012); Smith (2020)."

with \citep "...  with the frequency and severity of billion-dollar disasters steadily increasing over the past two decades (Campbell, 2012; Smith, 2020)"

I believe usually \citep{} is what \cite{} defaults to in natbib, but obviously I don't know the setup being used here.

----

It was said that the $(\lambda, \gamma)$ factors (and presumably also $\delta$??) were chosen by grid search, but it was not specified over what grid, nor how the model selection was performed.

**Audience:**

No

**Audience Explanation:**

In my opinion, as it is written, this submission falls outside the scope of this journal. Looking at the scope, the closest criterion within the ones outlined by the journal is "accounts of applications of existing techniques that shed light on the strengths and weaknesses of the methods". However, I think the insights about machine learning techniques that are shed by this work are not particularly deep, and instead greater insights about the structure of power outage data are gained (e.g., that it is important to enforce constrastivity between locations, or that the particular way of separating and integrating dynamic and static information is good). The techniques being employed are essentially adapted to these properties of the data, but are not shedding light on, e.g., alternative machine learning methods to address these particular concerns. Instead, these are shedding light on structural deficiencies in prior approaches to modeling this data.

Let me take careful stock of the proposed method to make this case. High level, there are three important pieces: 1) dynamic and static graph relationships should be modeled separately, and integrated only at the end, in this case by concatenation; 2) a contrastive loss can be used to handle the imbalances specific to outage data, and 3) the architecture incorporating these two modeling choices works well on power outage prediction.

Of these, 2, 3 are specifically claims about modeling power outages. This is certainly an important problem, but I don't know that the readers of a machine learning venue would find it particularly important from an ML perspective, and certainly the paper does not make the case that they would. 1. might be an interesting contribution form an ML perspective in that it makes a claim about how one should model certain dynamic dependencies, but it is not evaluated in sufficient generality for the case to be convincing outside of the power outage application. In particular, as I pointed out in the previous text box, I don't think that the evaluation is even sufficient to make the case that the observed effect on this data is robust to changing the architecture from using a GCN to another GNN method.

As such, in my opinion, this paper in its current state would find a more natural home in a venue dedicated to the study of power outages (or electrical engineering at large), rather than in a venue dedicated to machine learning. Indeed, one can see that bulk of the substantially discussed related work is published in such venues, whereas most of the related machine learning work is discussed in generic terms without precise contrasting with the proposed work.

**Broader Impact Concerns:**

No concerns

**Claims And Evidence:**

No

**Claims Explanation:**

Before offering subjective criticisms, I want to explicitly note that I am neither an expert in modeling of power outages, not in graph neural networks. As such, I have little ability to judge the novelty of the work in its proper context.

----




I think the paper suffers from muddled evaluation, and find that the writing is quite unclear in that even the problem is not completely specified in one place, but instead is drip fed to the reader.

- The evaluation is muddled. In particular, sectiion 5.1 does not seem to align well with the structure proposed, and appears to take the modular use of GCNs to be something essential to the architecture.

    I understand that the boosted decision trees are being compared to because prior power outage modeling work (in non-ML venues) studied these. These are not particularly compelling for this setting, although it is interesting that they often work very well. Similarly, a tabular that ignores the dependency structure is also not particularly compelling. My understand is that these comparisons show that attempting to incorporate the underlying graph into the predictions is a good idea.

    However, I do not understand how the competing graph neural networks are a valid comparison. Indeed, to my understanding, the contribution is a way of setting up the prediction model wherein the dynamic and static parts of the graph are both used to inform the representation. Here, the GCN forms just one part of the design, and as such it should be entirely possible to replace the use of the GCN in the architecture with any of the other GNN methods (like GAT or GIN or GraphSAGE as described in the paper).

    As such, then, it seems that the separated GCN architecture is being compared to unseparated GATs/GINs/GraphSAGES (henceforth OGNNs for "other GNNs"). But this is unnecessarily changing two things at once, and obscuring the effect of the contribution. If the claim is that the particular way of jointly incorporating the static and dynamic graphs is good, then it would be better to compare a GCN version of the SA-HGNN with a vanilla GCN, and similarly a OGNN version of SA-HGNN with a vanilla SA-HGNN. The current evidence does not thus actually tell us that the SA-HGNN method is a good idea in this problem, only that this with GCNs does better than OGNNs. This is further compounded by the fact that, presumably, the representations of the OGNNs are not supervised with the contrastive loss.

    Thus, in my opinion, the actual evaluation of the architecture proposed really lies in the ablations in Table 3, which I think should be the main result instead. Here the effect seen is good, but the question of how robust this effect is across architectures is left unanswered. I think a complete evaluation would reproduce something like ablation, with GCNs being changed for OGNNs.

-----

- The writing is quite unclear. In particular, it was not even clear to me what the actual structure of the available data is from the section describing the problem, and instead this is fed to us piecemeal through the paper.

    Let me expand upon this. Equation (1) states that $\hat{Y} = f\_{\textrm{GNN}}(X, A)$. However, the entirety of the rest of the paper is about the setting where this $A$ is not known, and must instead be estimated from the features $X$, and the procedure to do this is learned by fitting a model to such adjacency matrices learned during training. Thus, at the point of, say, section 4.1, one realizes that the learning setting is one where the training data is composed of triples of the form $(X\_k, A\_k, Y\_k)$, while at inference time only the features $X$ are available. However, even this is not quite true: in section 4.2 we learn that actually $X$ is composed of two pieces, a static channel $X^s$ that does not change across events, and a dynamic channel $X^d$ that does. Finally, and suddenly, instead of one graph, we have two graphs, a static one and a dynamic one.

    It's totally fine to have these subtleties in the model, but this should all be made clear in the problem description, rather than dripping out one by one.

---

Overall, while I think that the results of section 5.3 are interesting, the writing of the paper itself makes understanding the problem hard, and muddles up the evidence of the success of the design. Further, the results of this evaluation are not quite sufficient to establish that the architectural proposals are effective in a generic way even for the outage datasets.

**Requested Changes:**

If the authors do want to publish this work in an ML venue, I think it could be interesting if it were rewritten with a message of the following form, it would work better for this venue: "When modeling data with both dynamic and static dependencies, it is a good idea to separate out the static and dynamic parts in the bulk of the network, and only integrate them at the very end." This would be a substancial change, not an "adjustment": one would have to compare to other methods explicitly modeling dynamic dependencies (including the methods cited) and compare on data across many domains, rather than only power outage data.

---

> ### Author Response · Authors · 2026-01-09
>
> 1. It would be better to compare a GCN version of the SA-HGNN with a vanilla GCN, and similarly a OGNN version of SA-HGNN with a vanilla SA-HGNN.
> Thank you for this insightful comment. We have conducted additional experiments where we replace the GCN backbone within the SA-HGNN framework with alternative GNN architectures, including GAT, GIN, and GraphSAGE, while keeping all other components unchanged. The results show that SA-HGNN consistently outperforms its corresponding vanilla counterparts across architectures, and that the GCN-based instantiation achieves the best overall performance among the tested variants. These results demonstrate that the performance gains are from the proposed hybrid separation and dynamic graph learning framework rather than a specific backbone choice, while also indicating that GCN achieves the best performance in this setting.
>
>         Method                      AE q25        APE q25          MAPE
>         SA-HGNN (GCN)               25.00         23.09            52.77
>         SA-HGNN (GAT)               26.00         25.71            55.42
>         SA-HGNN (GIN)               28.00         23.23            53.83
>         SA-HGNN (GraphSAGE)         27.50         25.14            56.29
>
> 2. It was not even clear to me what the actual structure of the available data is from the section describing the problem.
> Thank you for this comment.  We provided detailed descriptions of the datasets in Section 5.1. To further clarify, our dataset is organized by service territory (CT, NH, WM, EM). For each territory, we observe m extreme weather events. In each event, a fixed set of n locations is monitored, and each location is represented by a fixed-length feature vector, yielding an event-wise tensor X_(m×n×k) . For a given event k, the input is Xk_(n×k). Importantly, while an event-specific adjacency matrix Ak  is available during training as external structural knowledge to guide learning, it is not assumed to be known at inference time. Instead, within our framework, the dynamic adjacency matrix is learned from the input features Xk  and inferred end-to-end for unseen events. The hybrid GCN design, which separates static and dynamic spatial dependencies and learns event-specific graph structure, is therefore a core methodological contribution rather than a prerequisite assumption of the problem setting. We acknowledge that this distinction between training-time guidance and test-time inference was not sufficiently explicit and have revised the problem statement to clearly present the data structure and learning setting while preserving the intended generality as shown in Section 3.
>
> 3. This submission falls outside the scope of this journal.
> Thank you for this thoughtful assessment. We respectfully disagree that the submission falls outside the scope of TMLR. First, power outage prediction under extreme weather is an important yet challenging real-world problem to tackle. Accurate forecasting of the magnitude and spatial distribution of weather-induced power outages can inform evacuation strategies, improve storm response planning, and guide investments in reinforcing and upgrading power infrastructure. Our aim in this paper is to develop innovative graph machine learning techniques to solve this problem. Specifically, we introduced SA-HGNN, a novel graph-based deep learning model that can effectively integrate both static and dynamic spatial dependencies to enhance outage prediction for extreme weather events. We developed a dynamic graph learning module that can capture and infer complex, evolving spatial relationships across different locations, addressing the limitations of existing methods that rely solely on fixed spatial structures. We also adopted a contrastive learning strategy that learns location-specific embeddings by minimizing intra-event distances between similar locations while maximizing inter-event distances across all locations. Our empirical studies on four utility service territories demonstrate that SA-HGNN can achieve state-of-the-art performance for power outage prediction. The problem studies in our paper arise broadly in spatiotemporal forecasting, infrastructure modeling, climate impact analysis, and are not specific to the outage domain.
>
> 4. The authors do not discuss the hyperparameters.
> Thank you for pointing this out. We have added a hyperparameter sensitivity analysis in Section 5.5 (Figure 7), where we systematically study the effects of the weighting coefficients lambda and gamma in Eq. 6. The results show that the proposed SA-HGNN is stable across a wide range of values and does not rely on fine-grained tuning.  We believe this addition addresses the concern and clarifies the role of hyperparameters in the model.

---

### Review · Reviewer_MRHa · 2025-12-17

**Summary Of Contributions:**

The authors introduce a novel GNN-based architecture designed to improve the prediction of power outages during extreme weather events. Their key contribution lies in better capturing the spatial dependencies between locations affected by such events. To do this, they propose a dual-channel embedding mechanism: one channel processes static features using a fixed adjacency matrix, while the other processes dynamic features using a learnable adjacency matrix that the model estimates during training.
The model is trained with a combination of contrastive loss, which encourages closer representations for positive intra- and inter-event node pairs, and regression objectives - i.e., estimate of dynamic adjacency matric; prediction of the actual number of outages. The authors evaluate the approach using simulated datasets based on four geographical regions. Their method outperforms existing approaches, with particularly strong improvements observed in the Connecticut case.

**Audience:**

Yes

**Audience Explanation:**

I find the idea of dynamically estimating the adjacency matrix particularly compelling for this application. Manually determining such relationships typically requires (i) detailed knowledge of the geographical and infrastructural context, and (ii) expert involvement to define and validate the relevant features. Your approach suggests a promising alternative: (i) use data from well-understood scenarios to (ii) train a model capable of inferring these spatial dependencies automatically, and (iii) deploy the trained model in new regions where such expert knowledge is not yet available.

To strengthen this contribution, the authors should elaborate further on this aspect by:
1) Clarifying the practical meaning of the learned adjacency matrix - for example, explaining how it relates to real-world phenomena such as how wind-induced failures might propagate from one location to another.
2) Motivating why automatic estimation is valuable in practice, including considerations such as cost, required expertise, and time constraints associated with manually constructing these matrices.
3) Show examples in which the knowledge learned for a set of locations "transfer" to others (at a different time, for different places, etc.)

Highlighting these points more explicitly would significantly enhance the practical relevance and impact of the proposed approach.

**Claims And Evidence:**

No

**Claims Explanation:**

The work is interesting and has clear potential. However, the current results appear somewhat limited: performance gains are particularly strong in the Connecticut case, but are only comparable to the baselines in the other regions. Since most of the subsequent analysis focuses on the Connecticut scenario, my main concern is: why are the improvements smaller in the other settings? Could this be related to the simulation parameters (e.g., a higher number of outages per event, which might amplify errors and thus affect performance)?

From Figure 2, when comparing your method to other approaches (e.g., TabPFN), it also seems that your model handles high-outage events better, while underestimating the distribution in cases with fewer outages. Is this interpretation correct?

More generally:
- It would be interesting to better understand how the model learns the different objectives. Have you monitored the loss components separately during training? What does the model learn first? Are the tasks learned jointly, or does the optimization plateau for one component while continuing for others? At the moment, the ablation study on the Connecticut dataset shows that all components contribute, but it does not reveal how the model uses the learned representations.
- The t-SNE visualizations are mainly qualitative. While Figure 3 is promising, I am not fully convinced that GAT behaves dramatically differently from your proposed method based on this alone.
- Similarly, Figure 5 provides useful intuition, but it would be more convincing to complement it with numerical metrics.
- As you already acknowledge, relying solely on simulated datasets is a significant limitation - especially when the simulation design itself may influence the behavior and apparent advantages of the proposed model.

**Requested Changes:**

- Clarify feature definitions with concrete examples (critical): Clearly distinguish static features (e.g., topography, presence of mountains) from dynamic features (e.g., wind speed at a given timestamp). Introducing examples early would significantly improve conceptual clarity.
- Strengthen motivation for learning a dynamic adjacency matrix (critical): Explain why manual construction is costly and often infeasible, and how learned adjacency supports generalization across regions (e.g., training in Connecticut, deploying in Western Massachusetts). Provide practical intuition for what the dynamic adjacency represents.
- Improve description of graph construction (Recommended): Clarify how nodes (locations) are grouped into events, the criteria used, and the typical graph size. How long these events last?
- Revise Figure 1 for clearer model flow (Recommended): Make explicit what the inputs and outputs of each module are. Consider separating the pipeline into phases (e.g., learning A^, creating embeddings, downstream prediction).
- Fix or clarify notation inconsistencies (critical): In Section 3, you use A to derive A(~). Should this instead be A^? If the model relies on A during training but not during testing, please clarify how inference works.
- Provide insights into learning dynamics (recommended): Report how different loss components evolve during training (e.g., adjacency learning, contrastive separation, outage regression). This would show what the model learns first and how tasks interact.
- Explain region-dependent performance differences (critical): The model performs best in Connecticut but shows marginal gains elsewhere. Explain why - e.g., simulation parameters, event severity distributions, or structural differences. This is central to evaluating robustness.
- Expand analysis beyond the Connecticut scenario (critical): Current qualitative analyses (e.g., adjacency visualizations, ablations) rely almost exclusively on the region where the model performs best. Provide at least partial analyses for another region.
- Add more detail on simulation procedure and limitations (critical): Provide stronger justification for lacking real-world validation or at least discuss challenges and planned future steps.

---

> ### Author Response · Authors · 2026-01-09
>
> 1. Performance gains are particularly strong in the Connecticut case and why are the improvements smaller in the other settings? Provide analyses for another region.
> We appreciate this careful observation about the larger performance gain in the CT region. This is primarily due to intrinsic data characteristics. The CT dataset contains a larger number of events and more severe storms with widespread outages, where spatial correlations among locations are stronger and more informative. In such settings, explicitly modeling event-specific spatial dependencies yields greater benefits. In other territories, many locations have few or no outage during the storm, resulting in weaker effective spatial signals. Thus, the advantage of complex graph-based modeling is less significant, leading to smaller performance gaps between SA-HGNN and baseline methods. Nevertheless, SA-HGNN remains consistently competitive performance across all regions. And we added a case study on EMA, demonstrating robustness across regions with different outage sparsity patterns.
>
> 2. It seems that your model handles high-outage events better, while underestimating the distribution in cases with fewer outages.
> Thank you for the insightful observation. Figure 2 shows our model more accurately captures high-outage events, while producing relatively conservative predictions for events with smaller outage counts. This behavior is intentional and aligns with the operational objectives of outage prediction. High-impact events contribute disproportionately to system risk and decision-making costs, and our model is designed to prioritize learning informative spatial and feature interactions in these regimes. In contrast, events with low outage counts often exhibit weaker spatial signals, where aggressive modeling can lead to overestimation rather than improved accuracy. Importantly, this trade-off does not degrade overall performance. As shown in Table 2, SA-HGNN consistently achieves the best or competitive results across standard quantitative metrics, indicating that it maintains strong accuracy across the full outage distribution while providing improved reliability for high-impact events.
>
> 3. Have you monitored the loss components separately during training?
> We monitored the individual loss components during training. In practice, the forecasting loss decreases rapidly in the early stages, indicating that the model first focuses on fitting the primary outage prediction task. As training progresses, the dynamic adjacency regularization stabilizes early and converges relatively quickly, while the contrastive objective decreases more gradually and continues to refine the learned node representations throughout training. This behavior suggests a natural learning progression in which the model first captures coarse predictive patterns, then incrementally improves the quality and structure of spatial representations.
>
> 4. Fig 5 would be more convincing to complement it with numerical metrics.
> Thank you for this suggestion. We have labeled actual and predict count in Fig 5. It is primarily intended to provide qualitative intuition. The case study focuses on a single representative event to illustrate spatial prediction patterns, while the quantitative evaluation is already provided comprehensively in Table 2, which reports performance across all events under the leave-one-out protocol.
>
> 5. The simulated datasets may influence the behavior and apparent advantages of the proposed model.
> Thank you for raising this concern. We would like to clarify that the datasets used in this work are NOT simulated outage datasets. As we discussed in Section 5.1, the outage datasets are collected from four real utility service territories, each consisting of hundreds to a thousand monitored locations which each location containing 390 features and outage counts.
>
> 6. Clarify feature definitions with concrete examples.
> Thank you for this suggestion.  In the revised manuscript, we introduced static features as time-invariant location attributes and dynamic features as event-specific meteorological variables early in the problem statement to improve conceptual clarity.
>
> 7. Strengthen motivation for learning a dynamic adjacency matrix.
> Thank you for this suggestion. We have strengthened the motivation in Section 4.1.
>
> 8. Fix or clarify notation inconsistencies.
> Thank you for pointing out this. We clarify that A(~) in the equation 3 denotes the adjacency matrix with added self-connections, following standard GCN notation, and is distinct from the learned event-specific adjacency A_k(^). And we have revised Section 4.1 to clarify the training and inference procedure: external adjacency information is used only during training to regularize the learning of A_k(^), while at inference time the adjacency matrix is inferred directly from input features without access to A. These clarifications resolve the notation inconsistency and make the inference process explicit.

---

### Comment · Action_Editor_u4nZ · 2026-01-02
**Rebuttal reminder**

Dear authors,

This is a reminder that you will have two weeks to submit rebuttals, since three reviews have been submitted.

Best wishes, AE

---

### Decision · Action_Editor_u4nZ · 2026-03-05

**Recommendation:** Accept with minor revision

**Additional Comments:**

- Citation format: should add a bracket () for in-line references.
- Related work: There is a different line of work on ensemble methods for GNNs [[1](https://dl.acm.org/doi/10.1145/3580305.3599265), [2](https://dl.acm.org/doi/10.1145/3637528.3671835)] and contrastive learning in imbalanced datasets [[3](https://arxiv.org/abs/2203.01517)]. Authors might consider discussing these in the related work section to strengthen connections to the machine learning literature.
- Eq (1): should it be $\hat A_k$ or $A_k$? In the previous paragraph, you said $A_k$.
- Code is not available. The authors should provide the code and data to replicate the empirical analysis (unless there are restrictions on data access, in which case they should be disclosed).
- Page 5: "Graph Convolutional Network Kipf & Welling (2016)" appeared in a hyperlink. Should be fixed.
- Table 1: Font size is too small.
- Section 5.1.3 Baseline Model: Remove hyperlinks?
- Further addressing reviewers' detailed comments before the final version, e.g., "The case studies are limited in scope and largely qualitative, while the quantitative metrics do not provide strong evidence in support of the claimed contributions. Notably, the proposed approach achieves substantially better results only in a single scenario (Connecticut). The authors argue that in the other cases the correlations between locations are weaker, and therefore there is little to be learned. This raises the question of the practical utility of the proposed methodology, given that in four out of five cases the results are comparable to existing approaches." And also addressing Reviewer QaSo's detailed comments.

**Audience:**

Yes

**Audience Explanation:**

Findings from this paper may be relevant to researchers studying computational sustainability and rare-event prediction using graph neural networks.

**Claims And Evidence:**

Yes

**Claims Explanation:**

This paper studies the problem of power outage prediction, which is a kind of rare-event prediction problem. The paper introduces Spatially Aware Hybrid Graph Neural Networks (SA-HGNN), which is a combination of spatial GNNs applied to the location network adjacency matrices, and contrastive learning for tackling the imbalanced data labeling in this problem.

The paper describes an empirical analysis conducted across four regions: Connecticut, Western Massachusetts, Eastern Massachusetts, and New Hampshire. This empirical study compares the above network architecture with other baselines, including ensemble methods (such as random forests) and graph neural networks.

The paper also presents ablation studies and additional case studies to further strengthen the empirical analysis above.

There are three detailed comments submitted for this paper. One main comment from Reviewer MRHa is that this paper is not considered a core topic in ML but rather an empirical study. While I agree with this comment, I believe this could be addressed by strengthening the discussion between the methodology component and other existing ML literature (see my detailed comments below). Another comment from Reviewer MRHa is that "the empirical evaluation is not fully convincing." For instance, I did not see the code or any guidelines for reproducing the empirical results in this paper. Another reviewer "recommended an Accept" after the authors had addressed their concerns during the rebuttal.

Taking all the information together, I recommend a minor revision for this paper (please find a list of more detailed editorial comments below).

---

> ### Author Response · Authors · 2026-04-03
> **Clarifications and Revisions**
>
> 1. Citation format: should add a bracket () for in-line references.
> Thank you for pointing this out. We have revised the citation format throughout the manuscript to ensure consistency with the required style, using parentheses for all in-line references.
>
> 2. Related work: There is a different line of work on ensemble methods for GNNs [1, 2] and contrastive learning in imbalanced datasets [3]. Authors might consider discussing these in the related work section to strengthen connections to the machine learning literature.
> Thank you for the suggestion. We have expanded the section 2 related work to cover graph-based multitask learning and contrastive learning, incorporating recent studies to better connect our work to the broader literature.
>
> 3. Eq (1): should it be $\hat A_k$ or $A_k$? In the previous paragraph, you said $A_k$.
> We have revised the notation throughout the manuscript to clearly distinguish between the observed adjacency matrix $A_k$, used as a structural prior during training, and the learned adjacency matrix $\hat A_k$, which represents the event-specific graph structure used for prediction.
>
> 4. Code is not available. The authors should provide the code and data to replicate the empirical analysis (unless there are restrictions on data access, in which case they should be disclosed).
> We appreciate the reviewer’s concern regarding reproducibility. The outage dataset used in this study is provided by Eversource and is proprietary, and therefore cannot be publicly released due to data sharing restrictions. However, we will release the code, synthetic datasets, and provide detailed implementation instructions to facilitate reproducibility upon request.
>
> 5. Page 5: "Graph Convolutional Network Kipf & Welling (2016)" and Section 5.1.3 Baseline Model appeared in a hyperlink. Should be fixed. Table 1: Font size is too small.
> We have removed unintended hyperlinks from the citation and ensured that all references are correctly formatted. Besides, we have changed the font size of Table 1 to improve readability.
>
> 6. Further addressing reviewers' detailed comments.
> We thank the reviewer for the insightful comment. We have carefully revised the experimental analysis to better clarify this point. First, we explicitly acknowledge in section 5.2 that the magnitude of improvement varies across regions, and we provide a more detailed explanation related to this variation to differences in spatial dependency structures. In particular, we observe that Connecticut shows stronger location correlations during extreme weather events, which allows spatially adaptive models such as SA-HGNN to provide larger gains. Second, we clarify that in regions with weaker or more localized spatial dependencies, the benefit of modeling long range spatial interactions is naturally reduced. Importantly, even in these cases, our method remains competitive with existing approaches and does not degrade performance, demonstrating robustness across diverse settings. Third, we strengthened the case study analysis by including quantitative evidence such as total outage prediction comparisons in addition to qualitative visualizations, providing stronger support for the effectiveness of the proposed method. Overall, the variation reflects the data characteristics, and SA-HGNN achieves strong improvements when spatial signals are informative while remaining stable in other cases.